# SARS-CoV-2 variant introduction following spring break travel and transmission mitigation strategies

Justin M. Napolitano[1,2], Sujata Srikanth[1], Rooksana E. Noorai[3], Stevin Wilson[3,4], Kaitlyn E. Williams[3,5], Ramses A. Rosales-Garcia[3,6], Brian Krueger[7], Chloe Emerson[1], Scott Parker[7], John Pruitt[7], Rachel Dango[7], Lax Iyer[7], Adib Shafi[7], Iromi Jayawardena[8], Christopher L. Parkinson[3,6], Christopher McMahan[9], Lior Rennert[8,10‡*], Congyue Annie Peng[1,11‡*], Delphine Dean[1,11‡*]

1 Clemson University, Research and Education in Disease Diagnostics and Intervention Clemson, Clemson, South Carolina, United States of America, 2 Department of Radiation Oncology, Wake Forest University School of Medicine, Winston-Salem, North Carolina, United States of America, 3 Clemson University, Clemson University Genomics and Bioinformatics Facility, Clemson, South Carolina, United States of America, 4 Illumina, San Diego, California, United States of America, 5 Clemson University, Center for Human Genetics, Greenwood, South Carolina, United States of America, 6 Department of Biological Sciences, Clemson University, Clemson, South Carolina, United States of America, 7 Labcorp, Burlington, North Carolina, United States of America, 8 Department of Public Health Sciences, Clemson University, Clemson, South Carolina, United States of America, 9 Clemson University, School of Mathematical and Statistical Sciences, Clemson, South Carolina, United States of America, 10 Clemson University, Center for Public Health Modeling and Response, Clemson, South Carolina, United States of America, 11 Department of Bioengineering, Clemson University, Clemson, South Carolina, United States of America

‡ These authors are joint senior authors on this work
* finou@clemson.edu (DD); congyup@clemson.edu (CAP); liorr@clemson.edu (LR)

**Data Availability Statement:** The sequence data underlying this article are available in the GenBank Nucleotide Database at https://www.ncbi.nlm.nih.gov/ and in the Global Initiative on Sharing Avian

## Abstract

### Background

University spring break carries a two-pronged SARS-CoV-2 variant transmission risk. Circulating variants from universities can spread to spring break destinations, and variants from spring break destinations can spread to universities and surrounding communities. Therefore, it is critical to implement SARS-CoV-2 variant surveillance and testing strategies to limit community spread before and after spring break to mitigate virus transmission and facilitate universities safely returning to in-person teaching.

### Methods

We examined the SARS-CoV-2 positivity rate and changes in variant lineages before and after the university spring break for two consecutive years. 155 samples were sequenced across four time periods: pre- and post-spring break 2021 and pre- and post-spring break 2022; following whole genome sequencing, samples were assigned clades. The clades were then paired with positivity and testing data from over 50,000 samples.

Influenza Data (GISAID) Resources at https://gisaid.org/. Accession numbers are provided in Supplementary Data Set 1.

**Funding:** This research received funding from multiple sources. SW, REN, CLP received support from Clemson University's College of Science. DD, CAP received support from Clemson University's Vice President for Research, Clemson University's Creative Inquiry, and the South Carolina Governor & Joint Bond Review Committee. SW, REN, CLP, LR, CAP, DD received funding through the National Institute of General Medical Sciences of the National Institutes of Health (https://www.nigms.nih.gov/; grant number: P20GM121342). SW, REN, CLP also received funding from the National Institute of General Medical Sciences of the NIH (under grant number P20GM109094). Additionally, CAP received support from NIGMS under grant P20GM139769. The funders had no role in the study design, data collection and analysis, decision to publish, or preparation of the manuscript.

**Competing interests:** I have read the journal's policy and the authors of this manuscript have the following competing interests: SW is currently employed by Illumina. BK, SP, JP, RD, LI, and AS are employed by Labcorp. The other authors declare no competing interest. This does not alter our adherence to PLOS ONE policies on sharing data and materials.

## Results

In 2021, the number of variants in the observed population increased from four to nine over spring break, with variants of concern being responsible for most of the cases; Alpha percent composition increased from 22.2% to 56.4%. In 2022, the number of clades in the population increased only from two to three, all of which were Omicron or a sub-lineage of Omicron. However, phylogenetic analysis showed the emergence of distantly related sublineages. 2022 saw a greater increase in positivity than 2021, which coincided with a milder mitigation strategy. Analysis of social media data provided insight into student travel destinations and how those travel events may have impacted spread.

## Conclusions

We show the role that repetitive testing can play in transmission mitigation, reducing community spread, and maintaining in-person education. We identified that distantly related lineages were brought to the area after spring break travel regardless of the presence of a dominant variant of concern.

## Introduction

In the spring of 2021, surging SARS-CoV-2 cases kept most universities from in-person education and adversely impacted the learning experience of college students. Implementation of a saliva-based RT-qPCR testing strategy has enabled several universities to retain in-person classes [1]. The high-complexity testing strategy allows surveillance testing of the entire university population with rapid turnaround time. However, institutions offering in-person instruction during the Spring 2021 semester faced a difficult decision regarding spring break. The risk of a spring break has two components: first, there is a hazard that students transmit the disease to their home communities or where they vacation, as well as any travel hubs along the way; second, there is a risk of students contracting COVID-19 during spring break and bringing it back to the university. A targeted surveillance strategy of students who traveled abroad during spring break documented these risks, while transmission risks to and from intermediate locations, such as airports and their surrounding regions, remain insufficiently understood [2,3].

In addition to intercommunal disease transmission, there are concerns regarding students returning from spring break and introducing new variants to the community. For example, Doyle et al. documented a clustered outbreak in an urban university in Chicago, caused by a variant new to the region, after the students returned from spring break [4]. A large, urban college campus in Pittsburgh implemented behavioral mitigation with targeted and random testing among the on-campus students and showed that a 0.4% positivity rate is maintainable with few outbreaks [5]. While successful, this strategy was only applied in an all-virtual learning environment. To strive for in-person learning and mitigation of SARS-CoV-2 spread in local communities with minimal outbreak clusters upon student return, we employed a comprehensive approach using saliva-based mass testing pre- and post-spring break. High-sensitivity saliva viral testing enables the capture of early asymptomatic infections in the university population so that proper isolation protocols can be established before travel [6]. Coupled with targeted whole genome sequencing, this strategy can identify variants with increased transmissibility and virulence introduced to the area after spring break travel. Swiftly relaying

these data to health authorities advises them on policies for mitigation and containment in the community.

Clemson University's Spring 2021 semester began in person on January 6th. All students with access to campus facilities were subject to pre-arrival and repeated weekly testing during in-person instruction using a saliva-based PCR test (testing protocols described in Ham et al.) [6]. The official 2021 spring break period was the week of March 15th. Upon return to campus, all students were required to undergo testing. Because residential students live in congregated settings more conducive to the spread of disease (Kasper, MR. 2020), a second COVID-19 test was required during the week of return (within 3–5 days after the first test) [7]. Starting in February 2022, the university moved to bi-weekly testing requirements; this policy was maintained before and during spring break and through the week following spring break; the official 2022 spring break period was the week of March 21st. The policy was changed to voluntary testing starting April 2nd, 2022; while the week after spring break (ending April 1st) was supposed to have maintained the mandatory testing policy, fewer students tested that week than expected. This testing, in addition to aiding in the mitigation of spread, also served as a source of material for numerous retrospective studies like this one.

This study provides a SARS-CoV-2 mitigation model fit for a college town where young, asymptomatic individuals with high mobility could contribute to highly infectious respiratory virus transmission. We show the effect of implementing SARS-CoV-2 mitigation strategies before students leave and after they return to campus. Results from this study capture the critical transmission of distantly related lineages introduced to the region. In the spring of 2021, the highly transmittable VOC, Alpha, became dominant after the students returned, indicating a VOC's introduction through student travel.

Under the dominance of the Omicron variant in Spring 2022, while no new transmission of VOCs to the area post-student travel occurred, phylogenetic analysis shows the introduction of distantly related Omicron sub-lineages to the community. Data presented in this study provides evidence of travel-related transmission of VOCs into the community where no apparent dominant lineage was present. Understanding the transmission dynamics of virus lineages associated with student travel will guide the university and community on preventative strategies to inhibit viral transmission.

## Materials and methods

### Saliva sample diagnostics

The Clemson University Research and Education in Disease Diagnosis and Intervention (REDDI) Lab (CLIA ID: 42D2193465) performed COVID-19 diagnostic tests from saliva samples from college students, employees and neighboring community members [1,6]. The Saliva-Direct™: RNA extraction-free SARS-CoV-2 diagnostics protocol Workflow 2 (Yale) was modified, with only one heat treatment step at 95˚C for thirty minutes instead of three [8]. Viral load was determined using RT-qPCR to measure the cycle threshold (Ct) value of the N1 gene. Ct Values equal to or less than 32 were determined to be COVID-positive, while samples with a Ct value above 32 were considered COVID-negative; the entire methodology is documented and shown as a video recording in by Ham et al. [6]. All samples were ethically collected and approved for use in the study described.

### Mitigation protocol and positivity rate study cohort

Data from the 2021 and 2022 spring break sample collection periods were analyzed in this study. The periods for analysis were as follows: pre-spring break 2021—March 1st to March 17th; post-spring break 2021 –March 20th to April 2nd; pre-spring break 2022 –March 5th to

March 19th; and post-spring break 2022 –March 26th to April 9th. In 2022, bi-weekly testing was mandated for two weeks before and one week after spring break. In this cohort study, we restricted the population to students tested in the two weeks before spring break who did not test positive before the study start date. We calculated weekly disease prevalence among students (residential and non-residential) as the ratio of unique positive COVID-19 cases to unique individuals tested. Differences in weekly prevalence were compared using risk ratios (RR) with Wald confidence intervals (CI).

## Whole genome sequencing study cohort and data generation

The periods for analysis were as follows: pre-spring break 2021—March 1st to March 17th; post-spring break 2021 –March 20th to April 2nd; pre-spring break 2022 –March 11th to March 25th; and post-spring break 2022 –March 26th to April 8th; note the 2022 periods for the positivity rate and whole genome sequencing studies are similar but not identical. This cohort includes 27 students from pre-spring break 2021, 101 from post-spring break 2021, 9 from pre-spring break 2022, and 18 from post-spring break 2022. A separate cohort for university employees and unaffiliated community members includes 47 (9 university, 38 community) from pre-spring break 2021 and 101 (41 university, 60 community) from post-spring break 2021; the university employee and unaffiliated community member 2022 cohort was small as very few people from this cohort tested positive during the time period of interest. For all cohorts, samples were deidentified and sequenced by the Clemson University Genomics and Bioinformatics Facility, along with three additional companies.

## Clemson University Genomics and Bioinformatics Facility (Clemson, SC)

RNA was extracted according to Zymo Research's Quick-DNA/RNA™ Viral MagBead kit protocol, including optional treatment steps. cDNA synthesis and library preparation were performed according to the COVIDSeq® Assay kit (Illumina) protocol. Libraries were pooled and sequenced on NextSeq® 550 (Illumina). Bcl2fastq version 2.17.1.14 (Illumina) was utilized to generate FASTQ files.

## Labcorp (Burlington, NC)

RNA was extracted using the MagMax™ MVPII kit (Thermo Fisher Scientific). cDNA synthesis and molecular inversion probe hybridization were performed according to the manufacturer's instructions (Molecular Loop Biosciences, Inc. ML5100). Synthesized DNA loops were amplified using index primers. Indexed libraries were pooled before being sequenced using the Illumina MiSeq® sequencing system. Standard data demultiplexing generated FASTQ files through the program NGmerge [9].

## Premier Medical Sciences (Greenville, SC)

RNA was extracted using the MagBind Viral RNA Kit (Omega Biotek, Norcross, GA) and was quantified via Logix Smart™ COVID-19 assay (Co-Diagnostics, Salt Lake City, UT). Samples were sequenced using the ARTIC protocol. Samples with sufficient RNA were sequenced on the Illumina platform according to the manufacturer's protocols.

## Aegis Sciences Corporation (Nashville, TN)

RNA was extracted using the MagMax™ Viral/Pathogen II Nucleic Acid Isolation Kit (Thermo Fischer Scientific). Library preparation was performed according to the COVIDSeq® RUO Reference Guide (# 1000000126053) workflow (Illumina), with minor modifications. Pooled

amplicons were sequenced on the NovaSeq™ 6000 (Illumina) according to the manufacturer's protocol. FASTQ files were generated with the FASTQ Generation v1.0.0 BSSH app.

## Clade and lineage prediction

All generated FASTQ files were aligned to the MN908947.3 reference genome and analyzed using nf-core/viralrecon v.2.4.1 [10]. Consensus sequences were generated for samples with at least 15,000 mapped reads and submitted to GenBank and Global Initiative on Sharing All Influenza Data (GISAID) (S1 Dataset). All samples had sufficient information to be confidently assigned a clade by Nextclade, with most also having a lineage predicted by Pangolin [11].

## Phylogenetic analysis

The university samples are represented by 155 consensus sequences. 3,158 SARS-CoV-2 sequences, including the reference genome (EPI_ISL_402125), were downloaded from the GISAID database deposited for North America between December 26th, 2019, and April 19th, 2022; these sequences were filtered using a 99% coverage threshold, leaving 2,188. 2,343 sequences (GISAID and university) were aligned using MAFFT [12]. The generated PHYLIP file was processed using IQ-TREE to infer a maximum-likelihood phylogeny [13]. Tree visualization and annotation were performed using FigTree version 1.4.4 (BEAST).

## Social media data collection and analysis

Social media data for this study were gathered using Sprinklr, a third-party platform capable of harvesting social media data from various platforms. Specifically, the search terms "spring break" OR "#springbreak" were used to collect posts from X (formerly Twitter) and Instagram across four separate periods (March through April for each year, 2019–2022).

## Ethics statement

The study protocol was reviewed and approved by Clemson University Institutional Review Boards (IRB) (IRB# 2021-043-02). This study is retrospective and uses bioreposited de-identified samples and data. Patient identifier information was removed before SARS-CoV-2 sequencing and data analysis. Informed consent was waived by the Clemson University IRB committee.

## Results

For the 2021 analysis, we included 12,607 unique students (20,583 tests) who tested in two weeks before spring break and 11,406 unique students (20,139 tests) who tested in the two weeks after spring break, per the mandatory testing policy; Fig 1 shows observed incidence (cases and rate) in this population. Prevalence was 0.5% in the week before spring break (3/6) and increased to 1.1% in the week following spring break (3/20), which is an increase by a factor of 2.12 (95% CI = 1.52–2.97, p < .001). Prevalence during the second-week post-spring break (3/27) and the week before spring break (3/6) did not significantly differ (RR = 1.36, p = 0.111). It should be noted that during the same time period, the number of positive tests from local community members did not significantly increase [19].

In 2022, bi-weekly testing was enforced before and during spring break but not after. In the analysis, we included 8,250 unique students (9,016 tests) who tested during the two weeks before spring break and 2,281 unique students (2,397 tests) who tested in the two weeks after spring break, per the mandatory testing policy. Fig 2 shows observed incidence (cases and rate); total student cases increased from 17 to 24, despite substantially fewer tests. While the

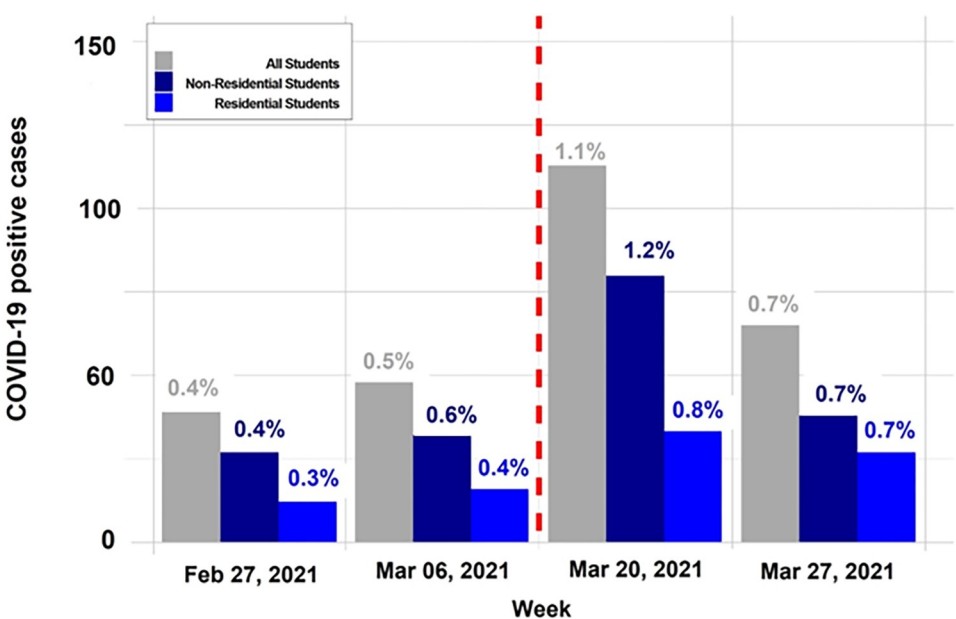

**Fig 1. 2021 spring break weekly COVID-19 cases and positivity.** Weekly COVID-19-positive cases and test positivity rates pre- and post-spring break in 2021; a mandatory testing policy was in place. COVID-19 cases and positivity rates of all students, non-residential students, and residential students are shown in gray, navy blue, and royal blue, respectively. The red dotted line indicates the post-spring break students' arrival to campus.

number of cases decreased in the second week following spring break, we do not have conclusive evidence to determine a causal effect of post-spring break testing due to a lack of mandatory testing in April. However, the previous 2021 showcases how post-spring break testing

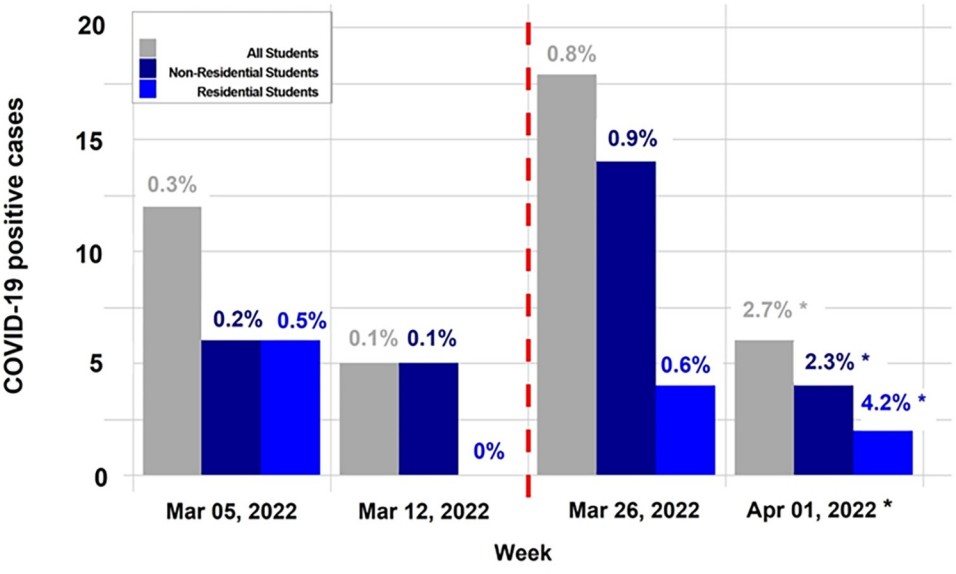

**Fig 2. 2022 spring break weekly COVID-19 cases and positivity.** Weekly COVID-19-positive cases and test positivity rates pre- and post-spring break in 2022; the mandatory bi-weekly testing was in place through the end of March 2022. COVID-19 cases and positivity rates of all students, non-residential students, and residential students are shown in gray, navy blue, and royal blue, respectively. The red dotted line indicates the post-spring break students' arrival on campus. *Because the testing mandate ended before April, test positivity rate does not represent prevalence.

**Table 1. Sample size, demographic information, and Ct values of the sequenced samples.**

| Node | Pre21 | Post 21 | Pre22 | Post22 |
|---|---|---|---|---|
| **Total** | **27** | **101** | **9** | **18** |
| **Age (years)** $\mu \pm \sigma$ | 20.8 ± 2.0 | 21.7 ± 2.9 | 22.2 ± 3.7 | 23.7 ± 5.0 |
| **Gender** | | | | |
| Male | 13 | 54 | 8 | 8 |
| Female | 14 | 47 | 1 | 10 |
| **Race** | | | | |
| Asian | 3 | 7 | 0 | 1 |
| Black or African American | 1 | 4 | 1 | 1 |
| White | 21 | 88 | 7 | 15 |
| Other | 2 | 2 | 1 | 1 |
| **N1 Ct (cycles)** $\mu \pm \sigma$ | 25.2 ± 3.4 | 23.4 ± 4.3 | 26.3 ± 4.6 | 27.6 ± 4.8 |

Age was calculated using the provided dates of birth. Ct values for individual tests were calculated by averaging two replicate Ct values.

reduced viral transmission. During the same four week time period, only one positive test out of 76 tests was reported from the community testing efforts.

We achieved SARS-CoV-2 variant surveillance through whole genome sequencing of the reposited samples from selected positive individuals. We observed the changes in the variant composition of positive SARS-CoV-2 samples from Clemson University and its surrounding community in upstate South Carolina (SC) before and after a spring break period of one week. Listed in Table 1 are the demographics and Ct values of the sequenced samples, including 128 from 2021 and 27 from 2022; note that the population is primarily white, with ages centered in the low 20s.

Fig 3 summarizes the clade assignment of the samples that passed the sequencing quality control. While we detected only four different clades before the 2021 spring break, the number of clades reached nine following spring break. In 2022, the emerging dominance of Omicron coincided with the detection of two strains before spring break and only three strains post-spring break. We found higher strain diversity after the 2021 spring break travel event than after the 2022 spring break travel event. We observed a shift of clade composition to a dominant VOC, including B.1.1.7 (Alpha), B.1.427 (Epsilon), B.1.429 (Epsilon), and P.1 (Gamma). The percent composition of Alpha increased from 22.2% to 56.4%, pre- to post-spring break, respectively. We observed an introduction of lineage 20J (Gamma) to the area post-spring break, which increased to 20.8% of the cases despite not being present pre-spring break. Similarly, Fig 4 shows that the population of university employees and unaffiliated community members tested also saw a large increase in the percent composition of Alpha (from 36.2% to 71.2%) and the introduction of Gamma (9.9%) and Epsilon (1.0%). VOCs with high transmissibility, such as Omicron, typically will have a significant advantage over and dominate other variants; this is shown in the 2022 student cohort when there was less clade differentiation when Omicron was present. While all the variants in 2022 were designated as Omicron by the World Health Organization, there was a shift in subclades within Omicron pre- and post-Spring Break, including the introduction of lineage 22C.

Table 2 details the relative frequency of words used in social media posts by Clemson University students from March through April of 2019, 2020, 2021, and 2022. 2020 and 2021 saw the introduction and increase in the frequency of words related to the COVID-19 pandemic, including "coronavirus", "positive", "test", "distancing", "covid", "cases", "pandemic", and "surge". These words saw a sharp decline in 2022 and were replaced with words of positivity,

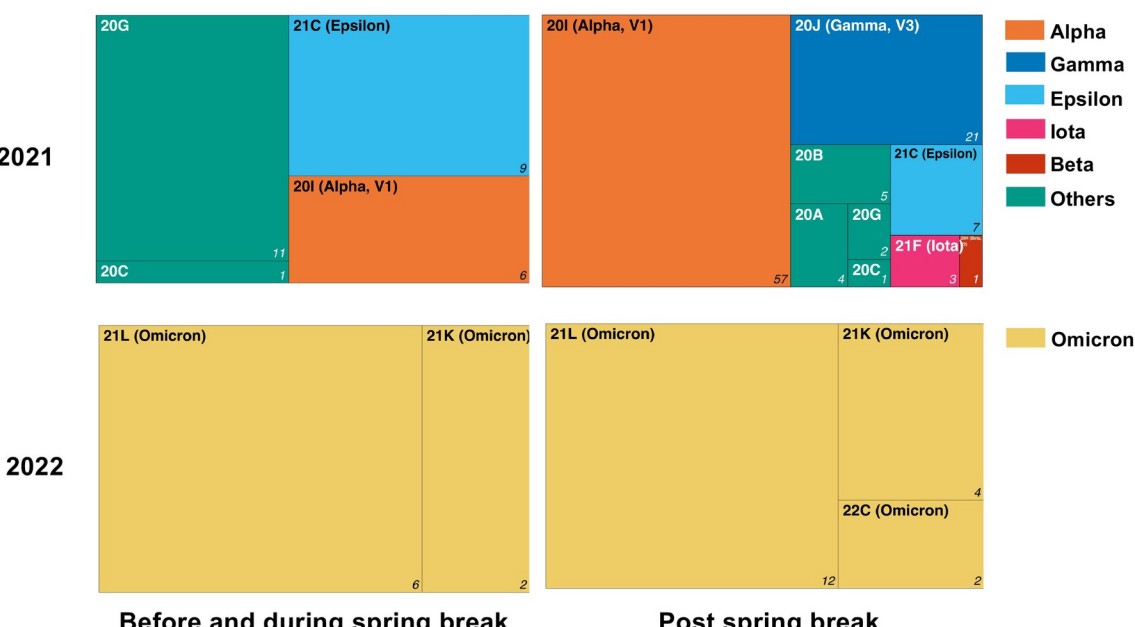

**Fig 3. COVID-19 clade assignment of the sequenced positive saliva samples during the study period.** The whole genome sequencing surveillance identified the 2021 and 2022 spring break transmission events. Generalized clade names are denoted by different colors, and the prevalence of each clade is indicated proportionally by the surface area of the block.

including "fun", "happy", and "enjoy". The frequency of words related to travel fluctuated throughout the four periods, though Miami, Florida, was consistently the most posted about the destination ("miami", "beach", "florida", and "trip"). S1 Fig shows word clouds for each year where the size of the words corresponds to their prevalence.

We further explored the lineage assignment and phylogenetic relationship of the collected samples. The phylogenetic tree, shown in Fig 5A, was built using the maximum likelihood method. Fig 5B shows a focused image of the branches highlighted by the black rectangles. In pre-spring break 2021, we primarily found lineages within nested clades, which indicates the presence of closely related strains. After the spring break of 2021, present lineages are found throughout multiple nodes and, in some cases, roots. This distribution indicates the increased presence of more distantly related isolates within the area. Similar to 2021, before and during

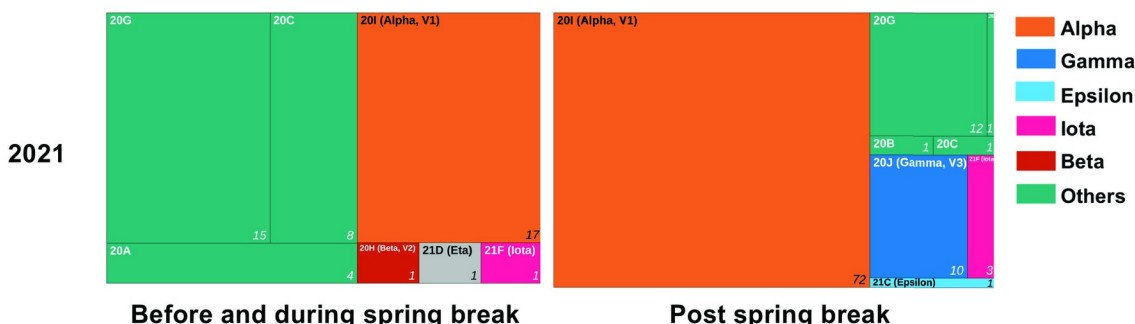

**Fig 4. COVID-19 clade assignment of the sequenced positive saliva samples of university employees and unaffiliated community members.** The whole genome sequencing surveillance identified the 2021 spring break transmission event. Generalized clade names are denoted by different colors, and the prevalence of each clade is indicated proportionally by the surface area of the block.

**Table 2. Ranked relative frequency of words regarding spring break, travel, the pandemic, and outlook used in social media posts before the pandemic (2019) and throughout the pandemic period (2020–2022).**

| Word | 2019 | | 2020 | | 2021 | | 2022 |
|---|---|---|---|---|---|---|---|
| | | | | Year | | | |
| Spring | 1 | | 1 | | 1 | | 1 |
| Break | 2 | | 2 | | 2 | | 2 |
| #springbreak | 3 | ↓ | NR | ↑ | 3 | ↓ | NR |
| Miami | 4 | ↓ | 8 | ↑ | 4 | ↓ | 5 |
| Friends | 5 | ↓ | NR | | NR | | NR |
| Beach | 6 | ↓ | 7 | ↑ | 5 | ↑ | 4 |
| Florida | 7 | ↑ | 3 | ↓ | NR | | NR |
| Trip | 8 | ↓ | NR | | NR | ↑ | 9 |
| College | 9 | ↓ | NR | ↑ | 7 | ↓ | 8 |
| Students | 10 | ↑ | 5 | ↓ | NR | | NR |
| coronavirus | NR | ↑ | 4 | ↓ | NR | | NR |
| Positive | NR | ↑ | 6 | ↓ | NR | | NR |
| Test | NR | ↑ | 9 | ↓ | NR | | NR |
| distancing | NR | ↑ | 10 | ↓ | NR | | NR |
| Covid | NR | | NR | ↑ | 6 | ↓ | NR |
| Cases | NR | | NR | ↑ | 8 | ↓ | NR |
| pandemic | NR | | NR | ↑ | 9 | ↓ | NR |
| Surge | NR | | NR | ↑ | 10 | ↓ | NR |
| School | NR | | NR | | NR | ↑ | 3 |
| Fun | NR | | NR | | NR | ↑ | 6 |
| Happy | NR | | NR | | NR | ↑ | 7 |
| Enjoy | NR | | NR | | NR | ↑ | 10 |

Words are ranked one through ten, where one is the most frequently used; words that were not in the top ten that year were denoted with NR (Not Ranked). Green up arrows indicate that a word was used relatively more frequently than the year before, while a red down arrow indicates that a word was used relatively less frequently than the year before.

the spring break of 2022, the strains detected are clustered to only a few branches, indicating they are closely related. After spring break, even though Omicron is the dominant circulating strain, the sub-lineages of different branches are further away from each other, indicating the introduction of distant strains. Our results showed that travel-related viral spread occurs with or without a dominant strain's presence. Under comparatively higher variant diversity in 2021, our data captured the travel-induced transition from the original strains to more virulent and highly transmissible VOCs. Whereas, under low variant diversity in 2022, data suggests the independent introduction of sub-lineages of the same variant to the region.

## Discussion

University campuses are conducive to disease transmission due to close living quarters and high-density social interactions among the adolescent population [1]. While the risk of severe illness from SARS-CoV-2 infection in the adolescent population is low, this population may disproportionately contribute to the spread of the disease compared to other age groups [14]. Ideally, during times of high infectious disease rates with public health concerns, such as a dangerous epidemic or pandemic, institutions should implement high-frequency repeated testing or alternative surveillance-based testing strategies for disease mitigation during the semester, though such strategies face increasingly obstructive barriers as they become more

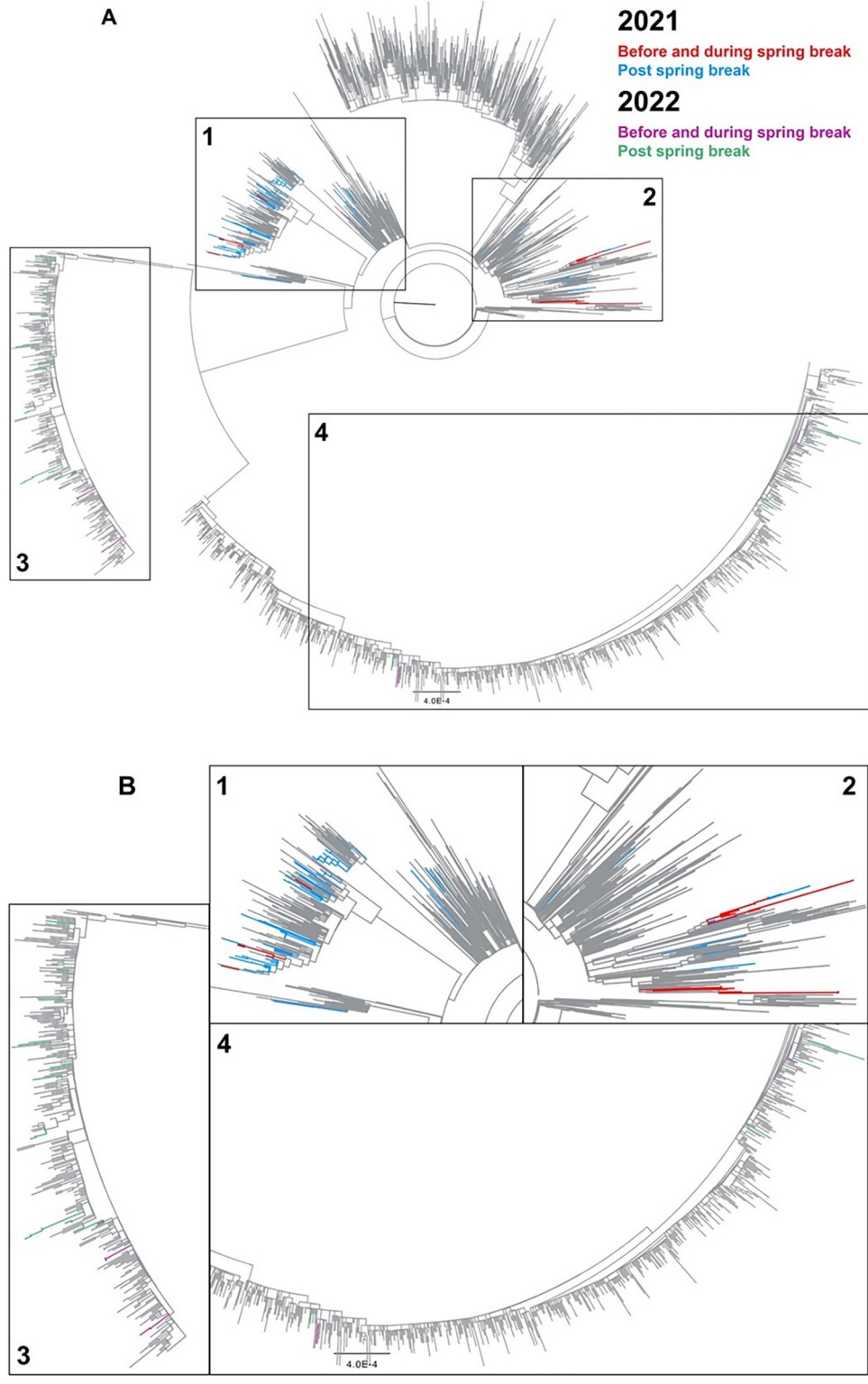

**Fig 5. Phylogenetic analysis of strains detected within the cohort.** Panel A) Phylogenetic analysis of the strains in our cohort during the study period related to the concurrent circulating strains in the United States. COVID-19 lineage was assigned using PANGO nomenclature. The sequences from GISAID are gray, while those from this study are colored based on their corresponding period. Before and during the 2021 spring break, clusters of lineages represent a few isolated events, and multiple isolates are descendants from the same lineage. Panel B) Zoomed-in

images of the areas indicated by the rectangles. Lineages identified in the pre- and post-spring break periods in 2021 are shown in red and blue, respectively. Lineages identified in the pre- and post-spring break period in 2022 are colored purple and green, respectively.

comprehensive [15–17]. At a minimum, institutions could require testing before students depart on travel events to minimize the risk of inter and intracommunal spread. By detecting COVID-19-positive cases before spring break, our study demonstrates that pre-departure weekly testing can substantially reduce the risk of transmitting university infection to outside communities. However, disease prevalence doubled in this population over spring break, indicating the need for return-to-campus testing post-travel. Failure to detect cases before campus return can lead to large outbreaks at institutions and surrounding communities [18]. In 2021, the university mandated a weekly testing policy throughout the entire testing period. In 2022 however, spring break marked the end of the mandatory testing policy; this policy change may pose a limitation to this report. External factors may have influenced COVID-19 prevalence, including community spread unrelated to the university [19]. While our study specifically focused on spring break, addressing future instances of individuals leaving and returning to campus en masse would be advantageous.

For both years, a spike in the positivity rate was detected immediately after spring break. Because of the vaccine rollouts in 2021, it is reasonable to argue that the spring 2022 positive rate may be affected by the protection from vaccination or previous infection. However, several studies conducted within the student population demonstrated that it is likely that a percentage of vaccinated and previously infected students still transmit the virus; discussion regarding the effectiveness of vaccines on Clemson University's population can be found in Rennert et al. [20–22]. Therefore, it is unlikely that vaccination and previous infection negatively affected the positivity rate, as we still saw an increased positivity rate after spring break.

VOCs have mutations that appear to increase transmissibility [23]. Alpha, Beta, and Gamma share an N501Y substitution, which increases the virus's binding affinity to the host's angiotensin-converting- enzyme-2 receptor. Studies suggest that increased binding affinity facilitates faster viral entry into cells, causing a higher viral load than variants without this mutation [24–26]. Our data support the high transmissibility of Alpha and Gamma, which became the leading representation of VOC post-spring break travel in 2021. Interestingly, despite finding 40% of all cases of the B.1.351 (Beta) variant detected in the US to be in SC during spring 2021, there was only one Beta sample found in our study [27]; this is surprising since many students did visit other parts of SC during spring break, before returning to campus. We expect that with a greater sample size, we may have seen a higher prevalence of Beta in the study. In addition, geographic patterns of the variants in the state may also explain the lack of Beta variants in our population. Figs 3 and 4 highlight the introduction of the Gamma variant to campus and the surrounding community after spring break, which may have come from other parts of the country as Gamma was not a high-prevalence variant in SC at the time [28]. Of note, the Gamma variant was prevalent in Florida at the time [28] which, consistent with the social medial analysis, may indicate that students did travel to that state. Spring break of 2022 occurred at the center point of the Omicron surge; the high transmissibility of Omicron dominated the infection cases. Due to the number of samples sequenced limiting the study, some circulating strains other than Omicron may have escaped detection. It is not surprising that Omicron surpassed every variant before and after the spring break in our data sets. However, phylogenetic analysis (Fig 5) demonstrated that although all cases were Omicron, the emergence of distantly related sub-lineages in a short time (within two weeks) indicates this is a travel-related transmission event.

It has been shown that domestic and international travel have both impacted transmission [29,30]. Analysis of social media trends (seen in Table 2) shows that Miami, Florida, was the most talked-about spring break destination of Clemson students before and after the start of the pandemic. Miami-Dade County saw between seven and eight thousand positive COVID cases reported (weekly) during Clemson's 2021 spring break and between 1,500 and 2,500 during Clemson's 2022 spring break; also note that in January 2022, Miami-Dade County saw an all-time high of nearly 110,000 weekly reported cases [31]. The county was a popular destination for college students outside of Clemson as well, so much so that emergency curfews and other measures were taken to manage the influx of students [32]. Travel to Miami and other destinations could have contributed to the introduction of new variants, seen in Fig 3; the Gamma variant, introduced to Clemson's population during the 2021 spring break, was responsible for nearly 3% of the sequenced cases in Florida and peaked in prevalence nationally approximately two months later [33]. While the social media analysis did not directly show if the quantity of spring break travel changed between years, it is notable that in 2022, words with positive associations increased in prevalence while no COVID-related language remained in the top ten. This could be reflective of students considering COVID less during the 2022 spring break compared to years before and may be tied to the significant increase in cases, as seen in Fig 2. All conclusions involving social media are limited by the reliability of the data itself; further investigation into social media trends could improve the understanding of students' behavior during the COVID-19 pandemic and future crises.

Phylogenetic inference based on this study and the whole genome sequences from GISAID also suggests that spring break travel resulted in strains of different geographical origins arriving in the region after the travel event. Thus, pre- and post-spring break mitigation methods appear critical to controlling spread, especially in groups not frequently surveilled, such as non-symptomatic young adults; such data should advise community response to future travel-related mitigation policies as additional VOC emerges. Due to evidence suggesting that college student travel significantly impacted local COVID-19 spread, many campuses adopted alternative semester break schedules [34,35]; evidence shows that shorter, more frequent breaks tend to be less detrimental to public health than longer, traditional breaks. Monte Carlo simulations confirm these results, and it would be reasonable to expect that rigorous surveillance testing (epidemiological control) would compound well with such strategies (administrative control) [35]. Using different types of controls in tandem has proven powerful for minimizing spread in educational settings, even in K-12 schooling, where students are too young to travel independently, and mass-testing is less feasible [36].

## Conclusions

We showed the benefits of repetitive testing, including mitigating transmission, reducing community spread, and maintaining in-person education. We concluded that spring breaks may lead to both increased viral spread and the introduction of new VOCs and sub-lineages to universities and surrounding communities; we hypothesize that spring break destinations could experience similar risks. We discovered a difference in students' views of the pandemic during the 2021 and 2022 spring break periods, which may have impacted transmission during these events. We identified that a dominant variant of concern's presence did not remove the risk of introducing new viral threats to a community, as was seen with the introduction of distantly related sub-lineages in 2022.

## Supporting information

**S1 Fig. Social media word clouds.** Four representations of the frequency of various words used during March and April of 2019, 2020, 2021, and 2022. Data was pulled from X (formerly

Twitter) and Instagram.
(TIF)

**S1 Dataset. Consensus sample sequences.** Generated for samples with at least 15,000 mapped reads and submitted to GenBank and Global Initiative on Sharing All Influenza Data (GISAID).
(XLSX)

## Acknowledgments

Clemson University is acknowledged for providing computing time on their high-performance computing resource, the Palmetto cluster. The Clemson University Social Media Listening Center is acknowledged for collecting and curating social media data. The authors would like to thank Rachel Ham and Kylie King of the REDDI Lab for technical support.

## Author Contributions

**Conceptualization:** Justin M. Napolitano, Stevin Wilson, Christopher L. Parkinson, Lior Rennert, Congyue Annie Peng, Delphine Dean.

**Data curation:** Justin M. Napolitano, Stevin Wilson, Ramses A. Rosales-Garcia, Lior Rennert, Congyue Annie Peng, Delphine Dean.

**Formal analysis:** Justin M. Napolitano, Sujata Srikanth, Rooksana E. Noorai, Stevin Wilson, Kaitlyn E. Williams, Ramses A. Rosales-Garcia, Iromi Jayawardena, Christopher McMahan, Lior Rennert, Congyue Annie Peng.

**Funding acquisition:** Rooksana E. Noorai, Stevin Wilson, Christopher L. Parkinson, Lior Rennert, Congyue Annie Peng, Delphine Dean.

**Investigation:** Justin M. Napolitano, Sujata Srikanth, Rooksana E. Noorai, Stevin Wilson, Kaitlyn E. Williams, Ramses A. Rosales-Garcia, Brian Krueger, Chloe Emerson, Scott Parker, John Pruitt, Rachel Dango, Lax Iyer, Adib Shafi, Iromi Jayawardena, Christopher L. Parkinson, Christopher McMahan, Lior Rennert, Congyue Annie Peng, Delphine Dean.

**Methodology:** Justin M. Napolitano, Rooksana E. Noorai, Stevin Wilson, Kaitlyn E. Williams, Iromi Jayawardena, Christopher L. Parkinson, Christopher McMahan, Lior Rennert, Congyue Annie Peng, Delphine Dean.

**Project administration:** Christopher L. Parkinson, Lior Rennert, Congyue Annie Peng, Delphine Dean.

**Resources:** Christopher L. Parkinson, Lior Rennert, Congyue Annie Peng, Delphine Dean.

**Software:** Justin M. Napolitano, Rooksana E. Noorai, Stevin Wilson, Ramses A. Rosales-Garcia, Iromi Jayawardena, Christopher McMahan, Lior Rennert.

**Supervision:** Christopher L. Parkinson, Lior Rennert, Congyue Annie Peng, Delphine Dean.

**Validation:** Justin M. Napolitano, Sujata Srikanth, Rooksana E. Noorai, Stevin Wilson, Kaitlyn E. Williams, Ramses A. Rosales-Garcia, Brian Krueger, Chloe Emerson, Scott Parker, John Pruitt, Rachel Dango, Lax Iyer, Adib Shafi, Iromi Jayawardena, Christopher L. Parkinson, Christopher McMahan, Lior Rennert, Congyue Annie Peng, Delphine Dean.

**Visualization:** Justin M. Napolitano, Sujata Srikanth, Rooksana E. Noorai, Stevin Wilson, Kaitlyn E. Williams, Ramses A. Rosales-Garcia, Lior Rennert, Delphine Dean.

**Writing – original draft:** Justin M. Napolitano, Sujata Srikanth, Rooksana E. Noorai, Stevin Wilson, Kaitlyn E. Williams, Chloe Emerson, Scott Parker, Christopher L. Parkinson, Lior Rennert, Congyue Annie Peng, Delphine Dean.

**Writing – review & editing:** Justin M. Napolitano, Sujata Srikanth, Rooksana E. Noorai, Stevin Wilson, Kaitlyn E. Williams, Brian Krueger, Chloe Emerson, John Pruitt, Rachel Dango, Lax Iyer, Adib Shafi, Iromi Jayawardena, Christopher L. Parkinson, Christopher McMahan, Lior Rennert, Congyue Annie Peng, Delphine Dean.

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
