## [Decision Letter · Decision Letter 0]

20 Dec 2023

PONE-D-23-32931SARS-CoV-2 variant introduction following spring break travel and transmission mitigation strategiesPLOS ONE

Dear Dr. Peng,

Thank you for submitting your manuscript to PLOS ONE. After careful consideration, we feel that it has merit but does not fully meet PLOS ONE’s publication criteria as it currently stands. Therefore, we invite you to submit a revised version of the manuscript that addresses the points raised during the review process. Your manuscript has been revised by two experts in t he field and both of thema raised concerns abiut this paper. In particualr one of the reviewer suggested to reject the manuscript due to major weakness points (see the report form the reviewers for details about the concerns found). If you feel that it could be possible to modify your work accordingly with tyhese many suggestions and consequently provide full response to the points raised, I would be happy to invite a revised version. Please submit your revised manuscript by Feb 03 2024 11:59PM. If you will need more time than this to complete your revisions, please reply to this message or contact the journal office at plosone@plos.org. Please include the following items when submitting your revised manuscript:A rebuttal letter that responds to each point raised by the academic editor and reviewer(s). You should upload this letter as a separate file labeled 'Response to Reviewers'.A marked-up copy of your manuscript that highlights changes made to the original version. You should upload this as a separate file labeled 'Revised Manuscript with Track Changes'.An unmarked version of your revised paper without tracked changes. You should upload this as a separate file labeled 'Manuscript'.

We look forward to receiving your revised manuscript.

Kind regards,

Vittorio Sambri, M.D., Ph.D.

Academic Editor

PLOS ONE

Journal Requirements:

"I have read the journal's policy and the authors of this manuscript have the following competing interests:

SW is currently employed by Illumina. BK, SP, JP, RD, LI, and AS are employed by Labcorp. 

The other authors declare no competing interest. "

Reviewers' comments:

Reviewer's Responses to Questions

**Comments to the Author**

1. Is the manuscript technically sound, and do the data support the conclusions?

Reviewer #1: Partly

Reviewer #2: No

2. Has the statistical analysis been performed appropriately and rigorously? 

Reviewer #1: I Don't Know

Reviewer #2: No

3. Have the authors made all data underlying the findings in their manuscript fully available?

Reviewer #1: Yes

Reviewer #2: Yes

4. Is the manuscript presented in an intelligible fashion and written in standard English?

Reviewer #1: Yes

Reviewer #2: Yes

5. Review Comments to the Author

Reviewer #1: The manuscript is interesting but still needs some work to improve its impact.

Napolitano and collaborators show the role that repetitive testing can play in transmission mitigation, reducing community spread of COVID-19 and identified that distantly related lineages were brought to the area after spring break travel of students.

In the ethic statement the Authors declare: This study is retrospective and uses bioreposited de-identified samples and data. Patient identifier information were removed before SARS-CoV-2 sequencing and data analysis. Informed consent was waived by the Clemson University IRB committee.

While at line 106-107 Authors state:

we employed a comprehensive approach using saliva-based mass testing pre- and post-spring break. Again at line 146-147 Upon return to campus, all students were required to undergo testing

So, it is not clear to me if the study was in fact retrospective or not, is samples collection part of this study or did the authors use bioreposited samples as declared in the ethics statement?

If this is the case, I would suggest to shift the text lines 142-165 in the introduction as sample collection was not part of the study.

The authors give some students’ demographic data such as age, gender and race, but it is not clear if these data have been very much analyzed e.g to compare groups is there any difference among groups in term of behavior, travel destinations etc

Besides some more details and discussion on the vaccination status of the students would be welcome (was this information available?)

Details and discussion about travelling destinations during the breaks would be needed.

Did the students travel only nationally or also internationally?

Did the Authors manage to collect information on COVID-19 VOCs from the areas visited by the students during the breaks? Is it possible to trace them back?

Impact of the paper:

The Authors conclude: We show the role that repetitive testing can play in transmission mitigation, reducing community spread, and maintaining in-person education.

It is not clear how testing can play in transmission mitigation and reducing community spread are the authors suggesting to keep on testing students before and after breaks? What kind of measure can be implemented in case they will test positive if there are no longer isolation policies in place?

“We identified that distantly related lineages were brought to the area after spring break travel regardless of the presence of a dominant variant of concern”.

Since it is quite well established that travelling is one of the most important factor increasing virus diversity, are the authors able to identify any suggestions/solutions (eg for surveillance purposes, for disease management and control etc) based on their data, to reduce the spread of SARS_CoV-2 and other respiratory viruses to be implemented at the level of mobile communities such as university students, and the academic community as a whole?

Reviewer #2: This work evaluates how students travelling during spring break may affect the university and surrounding communities, in terms of circulating variants and number of infections upon their return. The authors also discuss the role of repetitive testing as a mitigation strategy.

Considering the sample size and timeframe of the study, I cannot recommend the publication of this work. The role travelling plays in importing new variants into communities has been vastly described already, including students on spring break, as referenced by the authors themselves. The same can be said on the importance of repetitive testing, which has been evaluated in the past in a range of settings, from schools, to local labour force, to nursing homes.

Additionally, it is not clear whether the authors are suggesting coupling WGS with repetitive testing. While this certainly yields very useful information, it also slows down the turn around time considerably; this is crucial when advising health authorities on policies for mitigation and containment, especially when dealing with asymptomatic, highly mobile individuals.

Unfortunately this study does not add enough significant info to the body of work on the subject, at present.

6. PLOS authors have the option to publish the peer review history of their article (what does this mean?). If published, this will include your full peer review and any attached files.

Reviewer #1: **Yes: **Alessandra Scagliarini

Reviewer #2: No

---

## [Author Response · Author response to Decision Letter 0]

8 Mar 2024

Dear Editor 

Thank you for the feedback on our manuscript titled “SARS-CoV-2 variant introduction following spring break travel and transmission mitigation strategies”. We are grateful for the valuable comments provided by the reviewers, and each of the comments is addressed in this letter. To summarize, we updated the manuscript to meet PLOS ONE’s style requirement, we added the statement to confirm that the revision does not alter our adherence to all PLOS ONE policies on sharing data and materials, we added the full ethics statement in the ‘Methods’ section, we added additional methodology, data, and figures suggested by the reviewers. Here are our point by point responses: 

We updated our manuscript to comply with PLOS ONE’s style requirements. 

2. Please confirm that this does not alter your adherence to all PLOS ONE policies on sharing data and materials, by including the following statement: "This does not alter our adherence to PLOS ONE policies on sharing data and materials.” 

Here is the authors’ statement: 

“I have read the journal's policy and the authors of this manuscript have the following competing interests: SW is currently employed by Illumina. BK, SP, JP, RD, LI, and AS are employed by Labcorp. The other authors declare no competing interest. This does not alter our adherence to PLOS ONE policies on sharing data and materials.”

3. Please include your full ethics statement in the ‘Methods’ section of your manuscript file. In

your statement, please include the full name of the IRB or ethics committee who approved or

waived your study, as well as whether or not you obtained informed written or verbal consent. If consent was waived for your study, please include this information in your statement as well.

We added the following in the methods section:

The study protocol was reviewed and approved by Clemson University Institutional

Review Boards (IRB) (IRB# 2021-043-02). This study is retrospective and uses

bioreposited de-identified samples and data. Patient identifier information were

removed before SARS-CoV-2 sequencing and data analysis. Informed consent was

waived by the Clemson University IRB committee.

Review Comments to the Author

Reviewer #1: The manuscript is interesting but still needs some work to improve its impact.

Napolitano and collaborators show the role that repetitive testing can play in transmission

mitigation, reducing community spread of COVID-19 and identified that distantly related

lineages were brought to the area after spring break travel of students. In the ethic statement the Authors declare: This study is retrospective and uses bioreposited de-identified samples and data. Patient identifier information were removed before SARS-CoV-2 sequencing and data analysis. Informed consent was waived by the Clemson University IRB committee.

While at line 106-107 Authors state: we employed a comprehensive approach using saliva-based mass testing pre- and post-spring break. Again at line 146-147 Upon return to campus, all students were required to undergo testing. So, it is not clear to me if the study was in fact retrospective or not, is samples collection part of this study or did the authors use bioreposited samples as declared in the ethics statement? If this is the case, I would suggest to shift the text lines 142-165 in the introduction as sample collection was not part of the study.

Thanks for this comment. We moved the corresponding description of saliva sample collection and saliva testing to the introduction section. We also moved some information regarding 2022 sample collection to the same spot in the introduction as well. 

The authors give some students’ demographic data such as age, gender and race, but it is not

clear if these data have been very much analyzed e.g to compare groups is there any difference

among groups in term of behavior, travel destinations etc. Besides some more details and discussion on the vaccination status of the students would be welcome (was this information available?) 

This is a great comment. The data provided in this manuscript is a summary of the demographic information of the samples used for sequencing. We do not have any personal information on behavior and travel destinations as the samples were deidentified prior to sequencing and analysis. However, we did add analysis of social media data from the time period to get some average data on locations students went to for spring break. In addition, we analyzed tests from local community testing during the same time period as a comparison group. Local K-12 schools had different (later) spring break periods so that population was less likely to travel. Please See Table 2 and Supporting Fig 1.

We published a separate paper (cited below) detailing the discussions on vaccines and the omicron variant. We made additional comments in the discussion. 

Rennert L, Ma Z, McMahan CS, Dean D. Covid-19 vaccine effectiveness against general

SARS-CoV-2 infection from the omicron variant: A retrospective cohort study. PLOS Glob Public

Health. 2023;3(1):e0001111. doi: 10.1371/journal.pgph.0001111. Epub 2023 Jan 10. PMID:

36777314; PMCID: PMC9910751.

Details and discussion about travelling destinations during the breaks would be needed.

Did the students travel only nationally or also internationally?

This comment is very helpful. While we are unable to collect such information for individual patients as the individual samples were deidentified prior to analysis, we did do an additional analysis of social media for the period in question to get some information on student travel patterns for these spring breaks. Overall, from the social media analysis, the college student population mentions southeast beach destinations for travel (Florida and Miami were high resulting terms). There were no international destinations mentioned in the social media and international travel was also restricted especially during the 2021 spring break period so it is unlikely that this would contribute significantly the observed data. 

We added the information regarding the social media data, centering around a table (Table 2) we produced which ranked the prevalence of the words. We added the methodology for collecting this information in the method section. We added the word cloud figure in the supporting material (S1_Fig). 

Did the Authors manage to collect information on COVID-19 VOCs from the areas visited by

the students during the breaks? Is it possible to trace them back?

As mentioned above, while we are unable to collect the specific travel information of individual students, the social media analysis indicated that Florida (and Miami in particular) were likely to be popular destination. This is consistent with the VOC data recorded in the student population after spring break. Particularly in 2021, when new variants that had been circulating in Florida were found to be introduced in the student population after the spring break period. In addition, we have added the community sample sequenced data during that time frame for comparison. While the new VOCs did begin to circulate in that population the overall COVID rates remained consistent pre- and post-university spring break. 

Impact of the paper:

The Authors conclude: We show the role that repetitive testing can play in transmission

mitigation, reducing community spread, and maintaining in-person education.

It is not clear how testing can play in transmission mitigation and reducing community spread

are the authors suggesting to keep on testing students before and after breaks? What kind of

measure can be implemented in case they will test positive if there are no longer isolation

policies in place?

“We identified that distantly related lineages were brought to the area after spring break travel

regardless of the presence of a dominant variant of concern”.

Since it is quite well established that travelling is one of the most important factor increasing

virus diversity, are the authors able to identify any suggestions/solutions (eg for surveillance

purposes, for disease management and control etc) based on their data, to reduce the spread of SARS_CoV-2 and other respiratory viruses to be implemented at the level of mobile

communities such as university students, and the academic community as a whole?

Thank you for pointing this out. We agree with the reviewer that it is important that our data could be used to suggest mitigation management. We are addressing this by supporting both the restrictions and level of seriousness that we took in 2021 vs 2022. Essentially, people took COVID more seriously in 2021. This was also supported by the social media analysis which noted that in 2021 terms about spring break were associated with terms about COVID restrictions and mitigation measures. In contrast terms about spring break in 2022 showed very little association with COVID mitigation measures. Of note, while COVID rates increased after the spring break period in the Clemson student population, this did not seem to translate to an increase in COVID rates in the community. This may indicate that while students do certainly bring back more infectious diseases from travel during spring break, they are not likely to be a significant source of spread to the immediate non-academic population. It was not clear if this was due to the associated mitigation measures (isolation and quarantine) implemented during the pandemic or due to lower rates of mingling between the local community and university populations. 

Reviewer #2: This work evaluates how students travelling during spring break may affect the

university and surrounding communities, in terms of circulating variants and number of

infections upon their return. The authors also discuss the role of repetitive testing as a mitigation strategy. Considering the sample size and timeframe of the study, I cannot recommend the publication of this work. The role travelling plays in importing new variants into communities has been vastly described already, including students on spring break, as referenced by the authors themselves. The same can be said on the importance of repetitive testing, which has been evaluated in the past in a range of settings, from schools, to local labour force, to nursing homes. Additionally, it is not clear whether the authors are suggesting coupling WGS with repetitive testing. While this certainly yields very useful information, it also slows down the turn around time considerably; this is crucial when advising health authorities on policies for mitigation and containment, especially when dealing with asymptomatic, highly mobile individuals. Unfortunately this study does not add enough significant info to the body of work on the subject, at present. 

We understand the concerns the reviewer has about our study. We addressed the concerns in this revision to our best given the limitations with the data we could acquire. Due to the limitations of available sequencing resources, the sequenced samples are randomly selected from the testing samples collected from college students (N=27 and N=101 for pre- and post-spring break in 2021, respectively; N=9 and N=18 for pre- and post-spring break in 2022). We added a cohort for university employees and unaffiliated community members 47 (9 university, 38 community) from pre-spring break 2021 and 101 (41 university, 60 community) from post-spring break 2021 for comparison purposes. The number of sequenced samples from university employees and community members was small due to few patients testing positive with COVID during the associated time period. The data was included in the supplemental materials. In addition, while we are not able to know exact travel plans of individual student patients as the samples were deidentified, we added data on social media analysis to gain a qualitative understanding of where the students were likely to travel during the associated time period.

Although travel-related virus spreading has been documented and the importance of repetitive testing has been established, here, we are presenting a case study, that tracks a university population comprised of young individuals at low risk of hospitalization along the progression of the virus mutation during the pandemic and during the pandemic to endemic transition. We provide a model of the virus transmission in such a population using a highly calibrated phylogenetic tree assembly of a large amount of known virus lineage set ( > 2000 known virus lineage) in the midst of high virus gene pool diversity (year 2021) and low gene pool diversity (year 2022). The model has significance in both virus genetics and public health.

We agree with the reviewer that it is crucial when advising health authorities on policies for

mitigation and containment, especially when dealing with asymptomatic, highly mobile

individuals. The case we presented here provides an example of surveillance of the population

with testing in a large amount to slow down spreading, while using a randomly selected sample set for sequencing under resources and time limitations. We added social media data collection to strengthen our argument on the importance of testing and individual behaviors in virus mitigation.

---

## [Editor Report · Decision Letter 1]

13 Mar 2024

SARS-CoV-2 variant introduction following spring break travel and transmission mitigation strategies

PONE-D-23-32931R1

Dear Dr. Peng,

We’re pleased to inform you that your manuscript has been judged scientifically suitable for publication and will be formally accepted for publication once it meets all outstanding technical requirements.

Kind regards,

Vittorio Sambri, M.D., Ph.D.

Academic Editor

PLOS ONE
---

## [Editor Report · Acceptance letter]

26 Apr 2024

PONE-D-23-32931R1 

PLOS ONE

Dear Dr. Peng, 

I'm pleased to inform you that your manuscript has been deemed suitable for publication in PLOS ONE. Congratulations! Your manuscript is now being handed over to our production team.

Kind regards, 

on behalf of

Professor Vittorio Sambri 

Academic Editor

PLOS ONE